# Predictors of drug and substance abuse among school-going adolescents living in drug hotspot in Malaysia

**Rozmi Ismail**[1], **Nurul Shafini Shafurdin**[1]*, **Md Shafiin Shukor**[1],
**Azmawati Mohammed Nawi**[2], **Mohd Rizal Abdul Manaf**[2], **Norhayati Ibrahim**[3‡],
**Roziah Mohd Rasdi**[4‡], **Novel Anak Lyndon**[5‡], **Noh Amit**[3‡], **Siti Aishah Hassan**[4‡],
**Norshafizah Hanafi**[6‡], **Fauziah Ibrahim**[1‡], **Fathimath Nahla**[1], **Suzaily Wahab**[7‡]

1 Centre for Research in Psychology and Human Well-being, Faculty of Social Sciences and Humanities, Universiti Kebangsaan Malaysia, Bangi, Malaysia, 2 Department of Community Health, Universiti Kebangsaan Malaysia, Bangi, Malaysia, 3 Clinical Psychology and Behavioral Health Program, Faculty of Health Sciences, University Kebangsaan Malaysia, Bangi, Malaysia, 4 Universiti Putra Malaysia, UPM Serdang, Malaysia, 5 School of Development, Social and Environmental, Faculty of Social Sciences and Humanities, Universiti Kebangsaan Malaysia, Bangi, Malaysia, 6 Universiti Utara Malaysia, Bukit Kayu Hitam, Malaysia, 7 Faculty of Medicine, Universiti Kebangsaan Malaysia, Bangi, Malaysia

☯ These authors contributed equally to this work.
‡ NI, RMR, NAL, NA, SAH, NH, FI and SW also contributed equally to this work.
* nurulshafini@ukm.edu.my

**Data Availability Statement:** All relevant data are already in the manuscript.

## Abstract

This study explored the pressing issue of drug and substance abuse (DSA) among adolescents in drug hotspots in Malaysia. The Malaysian Anti-drug Agency (AADK) has identified 155 hotspot areas across Malaysia, of which 78 were randomly selected as the study sites. These areas were identified as having extreme drug-related activities such as drug trafficking and drug-related crimes. According to the literature, several factors influence adolescents to be involved in DSA. Therefore, understanding the risk factors in the context of Malaysian school-going adolescents is of utmost importance. The study examined, in particular, a wide range of potential predictors, including socioeconomic factors, peer influence, family dynamics, educational experiences, drug access, and community characteristics. Adolescents in the hotspot areas were selected by means of a cross-sectional survey design with a cluster sampling method. The sample comprised 3382 school-going adolescents, and the data were collected through face-to-face interviews. The logit model method with STATA software was used to analyse the data. The findings of the study revealed that school-going adolescents with disciplinary issues face a two-fold increase in the risk of becoming current drug users compared to their peers. Further, those exhibiting externalising behaviours, such as aggression and rule-breaking, also face greater odds of becoming involved in DSA. Drug pushers were identified as the most significant risk factor, with adolescents exposed to them being 46 times more likely to become current drug users. The factors of friends and family also contribute significantly to adolescent drug involvement. However, adolescents with academic-related issues may be less involved if they have the protective factor of better coping skills. These findings will contribute to efforts to mitigate drug addiction and drug-related activities, particularly in high-risk communities, as well as

**Funding:** The authors acknowledge the Ministry of Higher Education Malaysia and The Universiti Kebangsaan Malaysia, (UKM) for funding this study under the Long-Term Research Grant Scheme-(LGRS/1/2019/UKM-UKM/2/1). We also thank the research team for their commitment and tireless efforts in producing the manuscript. The funders had no role in study design, data collection and analysis, decision to publish, or preparation of the manuscript.

**Competing interests:** The authors have declared that no competing interests exist.

help policymakers and healthcare professionals develop targeted interventions and generally promote the well-being of adolescents.

## Introduction

Drug and substance abuse (DSA) among adolescents is a pervasive and complex public health issue [1] with significant implications for the well-being of individuals and society as a whole. In the Malaysian context, this challenge is further compounded by the existence of localised areas known as drug-based hotspots. These hotspots are characterised by alarmingly high rates of occurrence of drug-related problems among school-going adolescents, thus necessitating an in-depth exploration of the factors that contribute to this worrying phenomenon.

Past studies uncovered various elements that elevate the risk of drug involvement [2, 3] within different fields such as demographics, environmental stressors, peer pressure, family dynamics, and individual traits or characteristics [2]. Within the demographic factors, gender, age, and ethnicity have been found to be predictors of DSA. Globally, males, older adolescents, and those of Western ethnicity tend to report higher instances of DSA than other demographic groups. Similarly, in Malaysia, older age groups [4] and male adolescents have been found to be more at risk [5, 6] than their counterparts. In terms of race, *bumiputeras* (Malays, Sabahans, and Sarawakians) are more likely to engage in DSA, followed by Indians and Chinese [6, 7]. In addition, peer influence is a prominent and firmly established predictor of DSA among adolescents [8]. There is compelling evidence to indicate that unconventional attitudes and behaviours, such as tolerance of deviance, engagement in delinquent behaviour [9], and lack of knowledge of the dangers of drugs [10], correlate with DSA among adolescents. Living in high-risk areas also significantly increases the likelihood of adolescents participating in drug-related activities [8, 11].

Although the problem of adolescent DSA has been widely studied, research into creating diagnostic methods for recognising the psychological indicators of initial DSA is still inadequate [12]. The assessment of risk prediction has become crucial in the prevention of individual involvement in DSA. Understanding the factors that contribute to DSA and which are predictive of its occurrence is essential for the effective prioritization of intervention targets [9], especially in developing culture-based ones.

The investigation into the predictors of current DSA among school-going adolescents in the drug hotspots of Malaysia was underpinned by a synthesis of the pertinent theoretical frameworks in social sciences and public health. Grounded in the social learning theory [13], the study endeavoured to elucidate the mechanisms through which adolescents acquire attitudes and behaviours towards DSA by observing and imitating their social group [14, 15], encompassing peers [16–18] and familial influences [19].

Concurrently, the risk and protective factors framework served as an analytical scaffold to delineate the multifaceted interplay of individual, familial, peer, educational, and community-level determinants impinging upon the vulnerability or resilience of adolescents vis-à-vis DSA. Augmenting this inquiry, the ecological systems theory [20] was invoked to explore the nested ecological contexts encompassing microsystemic, mesosystemic, exosystemic, and macrosystemic influences, which collectively mould the drug-related behaviours of adolescents.

This study aimed to investigate the factors that can predict the occurrence of DSA among Malaysian school-going adolescents residing in drug-based hotspot areas. It is crucial to focus studies on hotspots to better understand their unique characteristics, the course of action, and

the right measures for effective preventive treatment. The efficacy of crime prevention strategies is augmented when directed towards identified hotspots and areas exhibiting heightened criminal activities [21]. A crime prevention study focusing on 110 identified hotspots led to a significant reduction in crime within eight months [22].

In Malaysia, a total of 155 drug-related hotspots were identified by the AADK in 2020. This overall number of hotspots was found to have decreased from the total of 178 hotspots reported in 2018. All 155 of these areas were located throughout Peninsular Malaysia, including Sabah and Sarawak. The breakdown of the hotspots was as follows: Perlis (six hotspots), Kedah (13), Pulau Pinang (seven), Perak (14), Selangor (13), Federal Territory of Kuala Lumpur (eight), Kelantan (15), Terengganu (15), Pahang (12), Melaka (10), Negeri Sembilan (six), Johor (17), Sarawak (seven), Sabah (nine), and the Federal Territory of Labuan (three). These hotspots were focal points that had been identified as grappling with drug addiction issues, with 60% of the entire nation's population under surveillance, while also serving as regions contributing to social issues such as social tourism and drug-related crimes such as robbery and drug trafficking. The designation of a hotspot is based on four primary indicators: the total number of individuals under surveillance, the total number of suspected individuals, the total number of new addicts, and the community readiness level. One of the strategies to minimise drug-related activities by 2025 is the "Hotspots Greening Programme" involving 155 hotspots in Malaysia, implemented collaboratively with government agencies and non-governmental organisations. The strategies of the programme are: a) preventing new drug addicts, b) maintaining recovery rates, c) increasing the detection of drug addicts, d) enhancing the cooperation and involvement of agencies or communities, and e) restricting drug entry.

The research elucidated several potential predictors comprising socio-economic factors, peer influence, family dynamics, educational experiences, drug access, and community characteristics. The purpose of considering this broad spectrum of variables was to construct a comprehensive and holistic picture of the factors that contribute to DSA among Malaysian school-going adolescents in hotspot areas. Through an in-depth understanding of these predictors, targeted and context-specific strategies can be developed to prevent and address DSA in these high-risk communities [23].

The significance of this research lies in its potential to address several critical issues. Firstly, it enables a nuanced understanding of the challenges faced by adolescents in drug-based hotspot areas, recognising that their experiences may differ subtly or significantly from those in other regions. Secondly, it provides a foundation for evidence-based policymaking and intervention design, ensuring that resources are allocated efficiently and effectively to combat DSA in the areas where it is most prevalent and crucial. Ultimately, the findings from this study hold the promise of making a substantial contribution to ongoing efforts to address DSA in Malaysia. By understanding the complex interplay of factors in these high-risk communities, this study aspires to empower policymakers, educators, and healthcare professionals with the knowledge needed to develop targeted interventions, reduce DSA rates, and promote the well-being and prospects of adolescents in these vulnerable areas.

## Methodology

### Research design, study area, and data collection

The current study was conducted between 1 December 2021 to 31 December 2022 using a cross-sectional survey design involving adolescents randomly selected from hotspot areas in Malaysia. These areas, characterised by alarmingly high rates of drug dealing and drug-related problems, have been gazetted by the National Anti-Drugs Agency (NADA). Face-to-face

interviews and questionnaires were administered by the trained researcher. Training was provided intensively to the research assistants by the project leader to ensure a good understanding of interview protocols, and interview simulations were also conducted. The students participating in the study ranged in age from 13–19 and came from 85 different schools under the Ministry of Education, representing five zones in Malaysia: North, East, Central, South and Malaysian Borneo (Sabah and Sarawak). The schools for the study were selected from the five zones covering eight states in total, namely Kedah, Penang, Kelantan, Terengganu, Selangor, Johor, Sabah, and Sarawak.

A total of 3382 school-going adolescents voluntarily participated in the study with the written consent of their parents or guardians. The inclusive criteria were Malaysian, school-goer, aged 13–19, parental consent given for those >18 years, no major physical or mental illness, and a good ability to write and read in Malay.

## Study instruments

The study consisted of sociodemographic, DSA, predictor, and other variables. The sociodemographic variables measured were state or place of origin, age, gender, ethnicity, years of residence, school environment, parents' relationship status, parent's occupation, and parent's income.

The study utilised a few instruments to measure DSA and its predictors. Drug and substance abuse (DSA) was measured using the Alcohol, Smoking, and Substance Involvement Screening Test (ASSIST) version 3.0 of the World Health Organization (WHO) 2010 [24]. The instrument measures lifetime and current DSA. Lifetime use is assessed using the question "*Have you used a drug in the past 30 days*?" with 10 categories of substances and drugs (cannabis, cocaine, amphetamine-type stimulants, inhalants, sedatives or sleeping pills, hallucinogens, opioids, and other drugs). Respondents have to answer "Yes" against the substance and drug categories to indicate if they have ever used any of the substances or drugs. Use over three months is considered as current DSA and is measured against the same 10 categories of substances and drugs using a 5-point Likert scale ranging from never (0), once or twice (2), monthly (3), weekly (4), daily or almost daily (6).

The predictor variables were measured using the Youth Self-Report (YSR) by Achenbach [25], while the independent variables were measured using Mooney and Gordon's problem checklist [26]. The YSR has 53 items, covering two domains, namely internal and external symptoms, along with four subdomains, namely withdrawn, anxious/depressed, aggressive, and rule-breaking behaviour, with a 5-point response scale ranging from strongly disagree (1) to strongly agree (5). The instrument was translated into Malay by Nik et al. [27]. Higher scores indicate adolescent emotional issues [28]. Mooney and Gordon's [26] problem checklist, validated by Ismail et al. [11], was used to examine problem areas, namely, financial, self-esteem, family, spiritual, educational, and future employment, with 10 items for each area. The questionnaire has 60 questions, and higher scores signify greater problems faced by adolescents in their respective aspects [29].

In addition, a Self-Report Coping Scale (SRCS) was used as a protective factor and identified as the independent variable. It was measured using problem-solving ability as one of the five scale ideas included in the SRCS to measure adolescent coping techniques. This study only examined the ability of adolescents to cope with academic- and peer-related issues using problem-solving, with eight items in each domain. Higher scores would indicate more regular usage of a coping mechanism [30].

The knowledge and attitudes (K&As) of adolescents toward DSA and sourcing of the drug, as independent variables, were also measured as predictors. To measure K&As, drug-related

knowledge, attitude, and belief scales [31] were used. The Malay version of the survey was used and adapted from Mahdi et al. [32]. This instrument is comprised of 35 items that exclusively assess drug-related K&As. Of these, the knowledge subscale has 16 items and the attitude subscale has 11 items, with the items being measured with a 7-point response scale ranging from strongly disagree (1) to strongly agree (7). Higher scores would indicate greater awareness of consequences, prevalence, treatment, prevention, and policy related to DSA. Thus, the respondents showed a greater appreciation and motivation to prevent community DSA and avoid being involved in drug-related activities. Lastly, this study also elucidated the sourcing of drugs, namely how drug and substance supplies were obtained. This part had three items related to the source of drugs; i.e., from friends, family, or pushers [3].

**Statistical analyses.** Selected variables were extracted from the raw data and displayed as frequencies and percentages for the categorical variables and as the mean (M) and standard deviation (SD) for the continuous variables. Pearson's $\chi2$ test was then conducted for the categorical and continuous variables to compare differences in current DSA and non-user abuse in relation to the independent variables. A bivariate analysis was conducted using IBM® Statistical Package for the Social Sciences (SPSS®) version 27.

A logistic regression analysis was performed using Stata version 15 to predict current DSA. The logit model analysis involved three stages. First, a simple logistic regression (SLR) was performed to assess the association between current DSA and sociodemographic factors, YSR measure, self-reported problems (SRP) measure, SRCS measure, K&As towards DSA, and source of drugs. Second, a multiple logistic regression (MLR) was used to assess the predictors of DSA, which was the significant variable of the SLR that was included in the analysis. Third, a stepwise logistic regression (SWLR) was used to assess the predictors of DSA after entering all the variables in the model. Only those variables with $p<0.05$ were retained in the final model. The adjusted odds ratios (ORs) and their 95% confidence intervals (CIs) were then estimated and associations with $p<0.05$ were considered statistically significant. All the variables in the study were adjusted by the dependent variable (current DSA) using the appropriate variance inflation factor (VIF) between the variables, and a VIF of $<5$ was used to detect multicollinearity. A Hosmer-Lemeshow test was used to measure the effectiveness of the model in describing the outcome variable. MLR and SWLR analyses were used. The pseudo-$R^2$ was used to measure the fitness of the model with the data. Correctly Classified was used to determine the extent to which the model could correctly classify or predict data; in other words, how well the model could accurately identify the correct data set. The β coefficients were estimated using the method of maximum likelihood.

**Ethics approval and informed consent.** The study was approved by the Ethics Committee of the Secretariat of Research Ethics, University Kebangsaan Malaysia, Cheras, Kuala Lumpur, with the reference number UKM PPI/111/8/JEP-2020-174(2). Written consent for the questionnaire survey was provided by each eligible participant, and the corresponding parental permission was obtained for those aged <18. Approval from the Education Planning and Research Division, Ministry of Education Malaysia, and state and district education offices was obtained before the data collection.

## Results

### Demographics characteristics and involvement in Drug and Substance Abuse (DSA)

A total of 3382 school-going adolescents aged 13–19 were sampled in the study. The characteristics of their involvement in DSA are shown in Table 1. Overall, 3.9% (n = 133) of the adolescents had never used drugs. In terms of locality, 4.3% (n = 70) of the 1625 adolescents in the

**Table 1. The general characteristics and extent of drug involvement of school-going adolescents in Malaysia (n = 3382).**

| Variable | Frequency | Percentage (%) | DSA Status Yes Frequency | DSA Status Yes Percentage (%) | DSA Status No Frequency | DSA Status No Percentage (%) | P* |
|---|---|---|---|---|---|---|---|
| **Locality:** | | | | | | | 0.313 |
| Rural | 1635 | 48.3 | 70 | 4.3 | 1565 | 95.7 | |
| Urban | 1747 | 51.7 | 63 | 3.6 | 1684 | 96.4 | |
| **Age:** | | | | | | | 0.056 |
| M | | 15.35 | | 15.83 | | 15.32 | |
| SD | | 2.279 | | 1.340 | | 2.322 | |
| **Gender:** | | | | | | | <0.001 |
| Male | 2425 | 71.7 | 121 | 5.0 | 2304 | 95.0 | |
| Female | 957 | 28.3 | 12 | 1.3 | 945 | 98.7 | |
| **Ethnicity:** | | | | | | | <0.001 |
| Malay | 2519 | 74.5 | 83 | 3.3 | 2436 | 96.7 | |
| Chinese | 261 | 7.7 | 12 | 4.6 | 249 | 95.4 | |
| Indian | 176 | 5.2 | 4 | 2.3 | 172 | 97.7 | |
| Bumiputera Sabahan or Sarawakian | 426 | 12.6 | 34 | 8.0 | 392 | 92.0 | |
| **Type of House:** | | | | | | | 0.391 |
| Landed property | 630 | 18.6 | 21 | 3.3 | 609 | 96.7 | |
| Shared property | 2752 | 81.4 | 112 | 4.1 | 2640 | 95.9 | |
| **Duration of Living in the Community:** | | | | | | | 0.046 |
| ≤9 years | 1027 | 30.4 | 30 | 2.9 | 997 | 97.1 | |
| ≥10 years | 2355 | 69.6 | 103 | 4.4 | 2252 | 95.6 | |
| **Positive Attitude towards School:** | | | | | | | 0.443 |
| Yes | 3180 | 94.0 | 123 | 3.9 | 3057 | 96.1 | |
| No | 202 | 6.0 | 10 | 5.0 | 192 | 95.0 | |
| **Changed Schools Due to Disciplinary Issues:** | | | | | | | <0.001 |
| Yes | 159 | 4.7 | 22 | 13.8 | 137 | 86.2 | |
| No | 3223 | 95.3 | 111 | 3.4 | 3112 | 96.6 | |
| **Parents' Relationship Status:** | | | | | | | 0.434 |
| Living together | 2709 | 80.1 | 103 | 3.8 | 2606 | 96.2 | |
| Separated | 673 | 19.9 | 30 | 4.5 | 643 | 95.5 | |
| **Father's Job Status:** | | | | | | | 0.098 |
| Employed | 2709 | 80.1 | 114 | 4.2 | 2595 | 95.8 | |
| Unemployed | 673 | 19.9 | 19 | 2.8 | 654 | 97.2 | |
| **Mother's Job Status:** | | | | | | | 0.805 |
| Employed | 1612 | 47.7 | 62 | 3.8 | 1550 | 96.2 | |
| Unemployed | 1770 | 52.3 | 71 | 4.0 | 1699 | 96.0 | |
| **Income Classification:** | | | | | | | 0.652 |
| B40 | 2909 | 86.0 | 118 | 4.1 | 2791 | 95.9 | |
| M40 | 384 | 11.4 | 12 | 3.1 | 372 | 96.9 | |
| T20 | 89 | 2.6 | 3 | 3.4 | 86 | 96.6 | |

Abbreviations: SD = standard deviation, B40 (<MYR4850/USD1155), M40 (MYR4850/ USD1155–MYR10970/USD2612, and T20 (>MYR10970/USD2612).
*Statistically significant if p<0.05.

rural areas admitted to DSA in the last 30 days prior to the survey. In comparison, only 3.6% (n = 63) of the 1747 urban adolescents admitted to the same. Males were more likely than females to be drug users (5.0%, n = 121) (n = 12: Female 1.3%). In terms of race, the largest

proportion of drug users were Bumiputeras from Sabah and Sarawak (8.0%), followed by Chinese (4.6%), Malay (3.3%), and Indians (3.2%). Most of the drug users were living in shared properties (4.1%, n = 112) and landed properties (n = 21:3.3%). Drug users tended to reside longer in their current community (4.4%, n = 103; with residence ≥10 years) compared to short-stayers (2.9%, n = 30). They also exhibited negative attitudes, i.e. disliked going to school compared to those who enjoyed doing so (5.0% and 3.9%, respectively). Most drug users had disciplinary issues (13.8%). In terms of parents' relationship status, most of their parents were separated (4.5%). Only 4.2% of their fathers and 3.8% of the mothers were employed. Most of the parents were classified as B40 (4.1%, n = 118). The bivariate analysis showed significant results (p<0.05) for the following variables; gender, race, duration of living in the community, and changing schools due to disciplinary issues.

**Social and behavioural characteristics.** The YSR scores indicated that most of the adolescents grappled with internal issues (M = 13.496; SD = 7.466), and external issues (M = 18.910; SD = 9.765) (Table 2). Therefore, most of them struggled with a range of emotional and behavioural challenges, both internally and externally.

The SRP scores indicated that the adolescents tended to prioritise their career goals (M = 35.128; SD = 8.539) (Table 2), followed by financial concerns (M = 34.346; SD = 5.739) and self-esteem-related issues (M = 33.226; SD = 6.940). Therefore, the adolescents faced stress and anxiety in their lives, particularly when confronted with concerns regarding career aspirations, financial stability, and self-esteem.

The SRCS scores indicated that most of the adolescents coped significantly better with peer-related issues (M = 26.707; SD = 7.341) than academic-related issues (M = 26.113; SD = 7.052) (Table 2). Therefore, they were more confident in managing social interactions and peer dynamics than academic issues.

**Table 2. The social and behavioural characteristics of school-going adolescents in Malaysia (n = 3382).**

| Variable | | | Drug Involvement Status | | | | P* |
|---|---|---|---|---|---|---|---|
| | | | Yes | | No | | |
| | M | SD | M | SD | M | SD | |
| **YSR:** | | | | | | | |
| Internal | 12.276 | 6.861 | 13.496 | 7.466 | 12.226 | 6.832 | 0.008 |
| External | 11.836 | 8.307 | 18.910 | 9.765 | 11.546 | 8.113 | <0.001 |
| **SRP:** | | | | | | | |
| Financial | 34.657 | 5.544 | 34.346 | 5.739 | 34.670 | 5.537 | 0.368 |
| Self-esteem | 33.201 | 6.446 | 33.226 | 6.940 | 33.200 | 6.426 | 0.192 |
| Family | 28.444 | 4.937 | 28.722 | 5.419 | 28.432 | 4.916 | 0.001 |
| Religious/moral issues | 33.974 | 7.290 | 32.842 | 8.286 | 34.021 | 7.243 | <0.001 |
| Job goals | 35.980 | 7.550 | 35.128 | 8.539 | 36.014 | 7.506 | 0.074 |
| Academics | 33.016 | 7.389 | 30.662 | 8.102 | 33.112 | 7.343 | <0.001 |
| **SRCS:** | | | | | | | |
| Ability to cope with academic-related issues | 27.655 | 7.011 | 26.113 | 7.052 | 27.718 | 7.003 | 0.576 |
| Ability to cope with peer-related issues | 28.415 | 7.397 | 26.707 | 7.341 | 28.485 | 7.392 | 0.245 |
| **K&As towards DSA:** | | | | | | | |
| Knowledge | 63.690 | 15.062 | 64.421 | 14.756 | 63.660 | 15.076 | 0.453 |
| Attitudes | 90.295 | 22.310 | 89.549 | 21.128 | 90.326 | 22.360 | 0.290 |

*Statistically significant if p<0.05.

In assessing their K&As toward DSA, the adolescents' attitudes were more significant (M = 86.549; SD = 21.129) than their knowledge (M = 64.421; SD = 14.756) (Table 2). Therefore, although they may have possessed relatively positive attitudes toward the issue of DSA, there was room for them to improve their knowledge and awareness of the subject.

In the bivariate analysis, several significant factors, such as the YSR, family problems, religious/moral issues, and academic-related concerns, were identified as significant (p<0.05). Thus, these factors are important in the various aspects of adolescent well-being and should be investigated further, and targeted interventions should be developed to mitigate the specific challenges.

## Source of drugs

There are three primary sources of drugs among school-going adolescents, as indicated in Table 3, namely friends, family members, and pushers (or dealers). The respondents largely obtained their current and lifetime drugs from friends and, to a lesser extent, from pushers/dealers and family members.

The notably greater importance of friends in drug sourcing signified the substantial influence of peer pressure and networking on the drug-related behaviours of adolescents. The role of friends as a source of drugs was particularly influential in this context. Additionally, the results underscored the importance of understanding and addressing peer dynamics when devising strategies to prevent and combat DSA among school-going adolescents.

**Predictors of Drug and Substance Abuse (DSA) among Malaysian school-going adolescents.** The SLR revealed that 12 predictor variables had an impact on DSA, either as risk factors or protective factors (Table 4). Of these 12 predictors, nine were associated with adolescents becoming current drug users. The risk factors included age, gender (male), those living in their communities for ≥10 years, those forced to change schools due to disciplinary issues, drug sourcing through friends, family, and pushers, and YSR from both internal and external sources. Additionally, the analysis identified three protective factors from the 12 predictors. These were excellence in academic performance, and a significant ability to cope with academic- and peer-related issues. These findings provided valuable insights into the factors influencing DSA among adolescents and emphasised the importance of addressing these risk and protective factors in univariate prevention and intervention efforts.

The multivariable analysis, as with the bivariate analysis, highlighted that an adolescent's engagement in other risky behaviours was strongly associated with current DSA (Tables 2 and 3). Table 5 The results of the MLR analysis indicate 11 predictors, 9 were associated with adolescents becoming current drug users. Meanwhile, the analysis identified three protective factors from current drug users

Table 6 indicates the results of the SWLR analysis that adolescents with disciplinary issues were twice as likely to become drug users (OR = 2.007; 95% CI: 0.379–0.911; p<0.05). Additionally, adolescents exhibiting externalising behaviours, such as aggression and rule-breaking, were more likely to become drug users (OR = 1.053; 95% CI: 1.031–1.075; p<0.01),

**Table 3. The sources from which school-going adolescents in Malaysia (n = 3382) obtained drugs.**

| Source | Drug Users (n = 133) | |
|---|---|---|
| | Frequency | Percentage (%) |
| Friends | 112 | 84.21 |
| Family | 6 | 4.51 |
| Pushers | 11 | 8.27 |

**Table 4. The results of the SLR analysis of the factors associated with DSA (n = 3382).**

| DSA Status (1 = Drug; 0 = Non-drug) | Coefficient | Unadjusted OR | Standard Error | Z-Score | Significance | 95% CI (OR) | |
|---|---|---|---|---|---|---|---|
| **Locality: (Reference: Rural)** | | | | | | | |
| Urban | -0.179 | 0.836 | 0.148 | -1.010 | 0.313 | 0.591 | 1.184 |
| **Age (Years)** | 0.190 | 1.209 | 0.101 | 2.280 | 0.023 | 1.027 | 1.423 |
| **Gender: (Reference: Female)** | | | | | | | |
| Male | 1.420 | 4.136 | 1.262 | 4.650 | <0.010 | 2.274 | 7.521 |
| **Type of House: (Reference: Landed property)** | | | | | | | |
| Shared property | 0.207 | 1.230 | 0.298 | 0.860 | 0.392 | 0.766 | 1.977 |
| **Duration of Living in the Community: (Reference: <9 years)** | | | | | | | |
| ≥10 years | 0.419 | 1.520 | 0.321 | 1.990 | 0.047 | 1.005 | 2.298 |
| **Positive Attitude towards School: (Reference: Yes)** | | | | | | | |
| No | 0.258 | 1.294 | 0.436 | 0.770 | 0.444 | 0.669 | 2.506 |
| **Changed Schools Due to Disciplinary Issues: (Reference: No)** | | | | | | | |
| Yes | 1.505 | 4.502 | 1.122 | 6.040 | <0.010 | 2.763 | 7.337 |
| **Parents' Relationship Status: (Reference: Living Together)** | | | | | | | |
| Separated | 0.166 | 1.180 | 0.250 | 0.780 | 0.434 | 0.779 | 1.789 |
| **Father's Job Status: (Reference: Unemployed)** | | | | | | | |
| Employed | 0.414 | 1.512 | 0.381 | 1.640 | 0.100 | 0.923 | 2.476 |
| **Mother's Job Status: (Reference: Unemployed)** | | | | | | | |
| Employed | -0.044 | 0.957 | 0.170 | -0.250 | 0.805 | 0.676 | 1.355 |
| **Income Classification: (Reference: T20)** | | | | | | | |
| Income B40 | 0.255 | 1.291 | 0.360 | 0.920 | 0.360 | 0.748 | 2.229 |
| Income M40 | -0.265 | 0.767 | 0.236 | -0.860 | 0.389 | 0.420 | 1.402 |
| **YSR:** | | | | | | | |
| Internal | 0.026 | 1.026 | 0.013 | 2.090 | 0.037 | 1.002 | 1.051 |
| External | 0.079 | 1.082 | 0.009 | 9.390 | <0.010 | 1.064 | 1.100 |
| **SRP:** | | | | | | | |
| Financial | -0.010 | 0.990 | 0.015 | -0.660 | 0.509 | 0.960 | 1.020 |
| Self-esteem | 0.001 | 1.001 | 0.014 | 0.050 | 0.964 | 0.974 | 1.028 |
| Family | 0.012 | 1.012 | 0.018 | 0.660 | 0.507 | 0.977 | 1.049 |
| Religious/moral issues | -0.022 | 0.978 | 0.012 | -1.830 | 0.068 | 0.956 | 1.002 |
| Job goals | -0.015 | 0.985 | 0.011 | -1.330 | 0.185 | 0.964 | 1.007 |
| Academic | -0.040 | 0.961 | 0.010 | -3.740 | <0.010 | 0.941 | 0.981 |
| **SRCS:** | | | | | | | |
| Ability to cope with academic-related issues | -0.031 | 0.970 | 0.012 | -2.580 | 0.010 | 0.947 | 0.993 |
| Ability to cope with peer-related issues | -0.030 | 0.970 | 0.011 | -2.710 | 0.007 | 0.949 | 0.992 |
| **K&As towards DSA:** | | | | | | | |
| Knowledge | 0.003 | 1.003 | 0.006 | 0.570 | 0.568 | 0.992 | 1.015 |
| Attitudes | -0.002 | 0.998 | 0.004 | -0.390 | 0.694 | 0.991 | 1.006 |
| **Source of Drugs:** | | | | | | | |
| Friends | 3.615 | 37.144 | 7.580 | 17.710 | <0.010 | 24.900 | 55.411 |
| Family | 2.390 | 10.917 | 5.418 | 4.820 | <0.010 | 4.127 | 28.875 |
| Pushers | 3.111 | 22.444 | 9.425 | 7.410 | <0.010 | 9.855 | 51.114 |

The most significant risk factor for adolescents becoming involved in drugs stemmed from their interactions with pushers (OR = 46.894; 95% CI: 18.748–117.295; p<0.01). Those exposed to their influence were 46 times more likely to become drug users, followed by friends

**Table 5. The results of the MLR analysis of the factors associated with DSA (n = 3382).**

| DSA Status (1 = Drug; 0 = Non-drug) | Coefficient | Adjusted OR | Standard Error | Z-Score | Significance | 95% CI (OR) | | VIF |
|---|---|---|---|---|---|---|---|---|
| **Age (years)** | -0.022 | 0.978 | 0.104 | -0.210 | 0.834 | 0.795 | 1.204 | 1.03 |
| **Gender: (Reference: Female)** | | | | | | | | |
| Male | 0.446 | 1.563 | 0.542 | 1.290 | 0.198 | 0.792 | 3.084 | 1.03 |
| **Duration of Living in the Community: (Reference: <9 years)** | | | | | | | | |
| ≥10 years | 0.357 | 1.429 | 0.363 | 1.410 | 0.159 | 0.869 | 2.350 | 1.02 |
| **Changed Schools Due to Disciplinary Issues: (Reference: No)** | | | | | | | | |
| Yes | 0.735 | 2.086 | 0.713 | 2.150 | 0.031 | 1.068 | 4.076 | 1.03 |
| **YSR:** | | | | | | | | |
| Internal | 0.030 | 1.031 | 0.019 | 1.690 | 0.092 | 0.995 | 1.068 | 1.46 |
| External | 0.039 | 1.040 | 0.013 | 3.160 | 0.002 | 1.015 | 1.066 | 1.42 |
| **SRP:** | | | | | | | | |
| Academic | -0.009 | 0.991 | 0.015 | -0.590 | 0.558 | 0.962 | 1.021 | 1.23 |
| **SRCS:** | | | | | | | | |
| Ability to cope with academic-related issues | -0.027 | 0.974 | 0.025 | -1.060 | 0.290 | 0.927 | 1.023 | 2.79 |
| Ability to cope with peer-related issues | -0.015 | 0.985 | 0.022 | -0.670 | 0.502 | 0.942 | 1.030 | 2.67 |
| **Source of Drugs:** | | | | | | | | |
| Friends | 3.619 | 37.306 | 9.111 | 14.820 | <0.010 | 23.116 | 60.209 | 1.06 |
| Family | 3.113 | 22.496 | 12.018 | 5.830 | <0.010 | 7.895 | 64.099 | 1.01 |
| Pushers | 3.871 | 47.999 | 23.347 | 7.960 | <0.010 | 18.501 | 124.528 | 1.02 |
| **Constant/Mean VIF** | -3.632 | 0.026 | 0.047 | -2.060 | 0.040 | 0.001 | 0.844 | 1.38 |
| Likelihood chi square (12) | 396.66 | | | | | | | |
| Significance | <0.010 | | | | | | | |
| Log likelihood | -359.6877 | | | | | | | |
| Pseudo R² | 0.3537 | | | | | | | |
| Hosmer-Lemeshow test-sig | 1.0000 | | | | | | | |
| Correctly Classified | 96.16% | | | | | | | |
| McFadden's R² | 0.354 | | | | | | | |
| Cox-Snell R² | 0.111 | | | | | | | |
| Nagelkerke R² | 0.392 | | | | | | | |

(OR = 38.069; 95% CI: 24.342–59.539; p<0.01), and family (OR = 23.800; 95% CI: 8.423–67.252; p<0.01).

Conversely, the SRCS identified a protective factor in that adolescents who exhibited coping skills related to academic issues were less likely to become drug users.

The pseudo $R^2$ of the model fit and performance was 0.3527, indicating that the variables in the study could explain 35.27% of the factors associated with DSA. The Hosmer-Lemeshow test (sig = 0.0314<0.010) suggested that the data fit the model, although a smaller p would have indicated a better fit. The Correctly Classified was 96.19%, thus, signifying that the model would be able to accurately predict whether a case belonged to the positive or negative class in approximately 96.19% of instances. McFadden's $R^2$ was 0.353, indicating that the model accounted for about 35.3% of the variation in the outcome variable. Meanwhile, the Cox-Snell $R^2$ was 0.110, suggesting that the model explained approximately 11.0% of the variation in the outcome variable. Lastly, the Nagelkerke $R^2$ was 0.391, indicating that the model accounted for around 39.1% of the variation in the outcome variable.

**Table 6. The results of the SWLR analysis of the factors associated with DSA (n = 3382).**

| DSA Status (1 = Drug; 0 = Non-drug) | Coefficient | Adjusted OR | Standard Error | Z-Score | Significance | 95% CI (OR) | | VIF |
|---|---|---|---|---|---|---|---|---|
| **Changed Schools Due to Disciplinary Issues: (Reference: No)** | | | | | | | | |
| Yes | 0.697 | 2.007 | 0.685 | 2.040 | 0.041 | 1.028 | 3.918 | 1.02 |
| **YSR:** | | | | | | | | |
| External | 0.051 | 1.053 | 0.011 | 4.790 | 0.000 | 1.031 | 1.075 | 1.04 |
| **SRCS:** | | | | | | | | |
| Ability to cope with academic-related issues | -0.042 | 0.959 | 0.015 | -2.740 | 0.006 | 0.931 | 0.988 | 1.01 |
| **Source of Drugs:** | | | | | | | | |
| Friends | 3.639 | 38.069 | 8.687 | 15.950 | 0.000 | 24.342 | 59.539 | 1.04 |
| Family | 3.170 | 23.800 | 12.614 | 5.980 | 0.000 | 8.423 | 67.252 | 1.00 |
| Pushers | 3.848 | 46.894 | 21.935 | 8.230 | 0.000 | 18.748 | 117.295 | 1.01 |
| **Constant/Mean VIF** | -2.738 | 0.065 | 0.035 | -5.100 | 0.000 | 0.023 | 0.185 | 1.02 |
| Likelihood chi-square (6) | 389.97 | | | | | | | |
| Significance | <0.010 | | | | | | | |
| Log-likelihood | -365.73437 | | | | | | | |
| Pseudo $R^2$ | 0.3477 | | | | | | | |
| Hosmer-Lemeshow test-sig | 0.0377 | | | | | | | |
| Correctly Classified | 96.16% | | | | | | | |
| McFadden's $R^2$ | 0.348 | | | | | | | |
| Cox-Snell $R^2$ | 0.109 | | | | | | | |
| Nagelkerke $R^2$ | 0.386 | | | | | | | |

## Discussion

The findings of this study highlighted that adolescents frequently encounter disciplinary issues that increase their likelihood of becoming current drug users. This phenomenon can be attributed to several factors. Firstly, adolescents are in a developmental stage, characterised by impulsiveness and a propensity for risk-taking behaviour [33]. Consequently, they may struggle with the self-control and discipline required to resist the allure of drug experimentation, particularly if driven by curiosity or the desire for novelty [34–36].

The study also revealed that male adolescents often exhibit externalising behaviours such as aggression and rule-breaking, which are strongly associated with an elevated risk of engaging in drug use [37–39]. These behaviours are interconnected with several risk factors. Adolescents, for example, may demonstrate aggressive or rule-breaking behaviours that gravitate them towards peers who share similar tendencies. Such peer networks may expose them to drug use that could rapidly undermine their decision-making and actions [8, 40]. Adolescents who display externalising behaviours may be more susceptible to peer pressure, feeling compelled to conform to the expectations and behaviours of their friends that revolve around drug use. Such naïve youngsters are prone to a heightened sensation-seeking personality trait that draws them to the excitement and risks inherent in drug use for the sheer novelty and stimulation effect [41].

The various sources of drug influence, namely, encompassing friends, family, and pushers, may play pivotal roles in shaping the drug-use patterns of adolescents. Friends, for example, can exert substantial peer pressure on adolescents, compelling them to conform to certain group behaviours, including drug use [42]. They may experience a strong desire to fit in with their peer group and be more inclined to experiment with substances if drug use happens to be at the core of their subculture. Additionally, immature adolescents frequently regard their new-found friends as role models and may quickly adopt their behaviours and attitudes,

including drug use, to gain peer acceptance [43]. Friends may also function as a direct source of drugs, introducing adolescents to substance abuse, since most of them are already hardened drug users or have connections with pushers [44].

Furthermore, social gatherings and recreational activities with such friends often provide a conducive environment for the normalisation and acceptance of drug use [45], thereby increasing the likelihood of experimentation by youngsters.

The family environment may also significantly influence the susceptibility of adolescents to drug use. Growing up, they often emulate the behaviour of family members, especially parents or older siblings [46]. If their family members happen to engage in drug use, this activity may naturally be perceived as the norm or as being acceptable within the household. Moreover, there may also be genetic predispositions to DSA within families, thus heightening the risk of adolescents developing drug-related addictions, especially if the family has a history of DSA [47].

Pushers also play an influential role in providing direct and easy access to drugs, thus posing a potent facility for impressionable adolescents to experiment with drugs. Pushers habitually operate within specific social networks or neighbourhoods, exposing adolescents to frequent drug-related activities and enticing them to participate. Youngsters from disadvantaged backgrounds, particularly those from low-income households, may be particularly susceptible to their influence. They are easily led to perceive drug distribution as an easy means to earn quick money or escape the challenges of poverty [48].

Adolescents with sufficient coping skills, however, are better protected from the attraction of drug use. As mentioned earlier, coping skills are related to academic problems and can reduce the likelihood of adolescents becoming current drug users. The relationship with drug use is, however, complex and may be influenced by various factors. Coping skills play a crucial role in enabling adolescents to manage stress, academic challenges, and other difficulties in life [49, 50]. Here are some insights into how effective coping skills may reduce the likelihood of adolescents turning to drugs.

Adolescents with strong coping skills may have healthier alternatives for dealing with academic study stress and challenges [51]. They may engage in activities like sports, arts, and various hobbies, or seek support from peers, mentors, or counsellors, rather than resort to drugs as a means of escape or stress relief. Effective coping skills can enhance resilience, thereby enabling adolescents to bounce back more effectively from school setbacks or various stresses [52]. Resilient individuals are less likely to turn to drugs as a way to escape or numb the emotional pain associated with study difficulties [53].

Additionally, adolescents with good coping skills, which often include emotional regulation techniques [54], tend to have better problem-solving abilities [55]. They are more likely to address academic challenges proactively, seeking help from teachers, tutors, or parents, rather than resorting to the use of drugs as a way to avoid or evade study issues. Conversely, they are less likely to turn to drugs to cope with negative feelings like anxiety, depression, or frustration resulting from academic problems.

Adolescents who have strong coping skills may also have better peer support networks [56]. Positive peer relationships can act as a protective factor against drug use as supportive friends may discourage or intervene when it comes to harmful behaviours that may be detrimental to the well-being of their members [8]. Likewise, adolescents with effective coping skills may have parents who are more engaged, caring, and supportive. Such parents are often better equipped to recognise signs of distress in their children and offer guidance and assistance, thereby reducing the likelihood of DSA [57].

Furthermore, adolescents who possess coping skills related to academic study problems may also be better informed about the risks associated with drug use. This special knowledge

can act as a deterrent since they understand the potential negative consequences of DSA [58]. Effective coping skills can lead to better academic performance, which in turn, can boost self-esteem and self-efficacy. Adolescents who excel academically are less likely to engage in drug use as they value their achievements and are motivated to maintain their success.

## Policy implications and study limitations

The policy implications of the coping skills of adolescents in relation to their academic performance should be emphasised.

Firstly, educational institutions and the relevant stakeholders must prioritise the integration of coping skills training programmes into the school curriculum. These programmes can assist adolescents in developing effective coping mechanisms to manage stress and challenges related to their schooling. The relevant initiatives that can be taken may encompass stress management workshops, emotional regulation training, and the development of problem-solving skills.

Secondly, it is crucial to encourage the formation of positive and supportive peer networks within schools. Teachers can implement peer mentoring programmes or establish peer support groups to facilitate interactions among students. These supportive relationships can serve as a protective barrier against drug use by promoting healthier ways of dealing with stress and fostering a sense of belonging and understanding.

Thirdly, parents play a vital role in nurturing coping skills and motivation for academic success in their children [59]. Policymakers and schools should collaborate to provide resources and workshops for parents to enhance their understanding of adolescent development and effective parenting strategies. Such support can empower parents to recognise the signs of distress in their children and provide appropriate assistance to mitigate their predicaments.

Fourthly, public health campaigns should be designed to educate adolescents on the risks associated with drug use. These campaigns should target both adolescents and their parents in order to emphasise the importance of maintaining good academic performance and highlight the potential negative consequences of DSA on academic success and future prospects.

Fifthly, schools should implement timely interventions for at-risk students, who characteristically exhibit poor coping skills or declining academic performance. These initiatives may include the provision of counselling services, academic support programmes, and referrals to external resources. Identifying and addressing the early onset of academic and emotional challenges may prevent students from resorting to drugs as a coping mechanism.

Lastly, policymakers need to establish mechanisms for monitoring and evaluating the effectiveness of coping skills programmes and interventions implemented in schools. Regular assessments can help identify areas for improvement and ensure that resources are allocated efficiently to programmes that yield positive results.

Emphasising these policy implications can effectively contribute to the prevention of DSA among adolescents. Such interventions will strengthen the coping skills and academic performance of susceptible youngsters and ultimately, foster a healthier and more promising future for them.

The main strength of the study is that it is a nationwide study based on hotpots. It also provides an insightful understanding of drug-using adolescents in high-risk areas. However, in interpreting the results of the study some limitations need to be taken into account. The survey was targeted at adolescents aged 13–18. Since DSA is considered a taboo and sensitive issue among them as well as the larger community, some of the respondents may have been hesitant to provide accurate information in order not to reveal their past experiences. These individuals

may have encountered difficulties in truthfully communicating their responses, particularly on past drug-related events and activities. Moreover, some of these respondents may, unsurprisingly, have continued to harbour feelings of fear and suspicion, since any truthful disclosure of information might be detrimental to their future. Hence, the SRP data may have been subject to respondent bias. Perhaps a more reliable and precise source of information on DSA could have been obtained by employing an approach that could effectively reach out and engage with segments of the adolescent population. These potentially more suitable sources, such as school dropouts, are typically well-concealed and challenging to access and are not necessarily found in the hotspot areas alone.

## Conclusion

In conclusion, by prioritising the integration of coping skills training programmes in educational institutions and other initiatives, as suggested, such as fostering positive peer networks, empowering parents with knowledge and support, conducting effective public health campaigns, implementing targeted interventions, and establishing mechanisms for programme evaluation, policymakers and stakeholders may work together to prevent DSA among adolescents. These findings can be utilised to strengthen the existing policy measures for adolescents to initiate a drug-free community. These main implications of the study include strengthening awareness, early identification, and intervention for adolescents and parents through universal school-based programs and targeted programs. A universal program such as social emotional learning programs can be introduced into school curriculum and extracurricular activities from early adolescence onwards which involves learning emotion management, healthy coping skills, and interspersal skills.

Drugs and substance harm awareness programs can be aimed for middle adolescents and parents. Additionally, strengthening the current ongoing random screening system that is only limited to urine tests can be supplemented with adding a layer to school-based screening for identifying students with emotion management difficulty and interpersonal skill deficit. Furthermore, implementing targeted programs such as skills groups and individual sessions for at risk adolescents through school counsellors and a referral process along with targeted interventions for adolescents involved in substance with the help of out-patient experts can be implemented.

Additionally, engaging adolescents in school to enhance academic and vocational training can be beneficial. These interventions will help in building individual psychological capital, to equip adolescents with skills to deal with social environmental influences, and early identification of at-risk individual to provide support with and a layer of support for individuals involving in substance use. Therefore, these can ultimately pave way for healthier and a more promising future for our youth. The strengths of the study include nationwide data and discovery of useful knowledge on the current situation. Nevertheless, the data were self-reported and focusing on school-going adolescent may potentially leave out the high-risk adolescents who have dropped out of schools.

## Author Contributions

**Conceptualization:** Rozmi Ismail, Nurul Shafini Shafurdin, Md Shafiin Shukor, Azmawati Mohammed Nawi, Mohd Rizal Abdul Manaf, Norhayati Ibrahim, Roziah Mohd Rasdi.

**Data curation:** Nurul Shafini Shafurdin, Md Shafiin Shukor, Mohd Rizal Abdul Manaf, Fathimath Nahla.

**Formal analysis:** Md Shafiin Shukor.

**Funding acquisition:** Rozmi Ismail.

**Methodology:** Rozmi Ismail, Nurul Shafini Shafurdin, Md Shafiin Shukor, Azmawati Mohammed Nawi, Mohd Rizal Abdul Manaf, Norhayati Ibrahim.

**Project administration:** Rozmi Ismail, Norhayati Ibrahim.

**Supervision:** Rozmi Ismail, Suzaily Wahab.

**Writing – original draft:** Md Shafiin Shukor.

**Writing – review & editing:** Nurul Shafini Shafurdin, Md Shafiin Shukor, Azmawati Mohammed Nawi, Mohd Rizal Abdul Manaf, Norhayati Ibrahim, Roziah Mohd Rasdi, Novel Anak Lyndon, Noh Amit, Siti Aishah Hassan, Norshafizah Hanafi, Fauziah Ibrahim, Fathimath Nahla.

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
