## [Decision Letter · Decision Letter 0]

6 Mar 2024

PONE-D-23-40678PLOS ONE Subject: Submission of Manuscript entitled Predictors of Current Drug Use among the School-going Adolescents in the Drug Hotspots of Malaysia.PLOS ONE

Dear Dr. SHAFURDIN,

Thank you for submitting your manuscript to PLOS ONE. After careful consideration, we feel that it has merit but does not fully meet PLOS ONE’s publication criteria as it currently stands. Therefore, we invite you to submit a revised version of the manuscript that addresses the points raised during the review process.

We look forward to receiving your revised manuscript.

Kind regards,

Assoc Prof Dr Nik Ahmad Sufian Burhan

Academic Editor

PLOS ONE

Journal Requirements:

The authors acknowledge the Ministry of Higher Education Malaysia and The Universiti Kebangsaan Malaysia, (UKM) for funding this study under the Long-Term Research Grant Scheme-(LGRS/1/2019/UKM-UKM/2/1). We also thank the research team for their commitment and tireless efforts in producing the manuscript.

The authors acknowledge the Ministry of Higher Education Malaysia and The Universiti Kebangsaan Malaysia, (UKM) for funding this study under the Long-Term Research Grant Scheme-(LGRS/1/2019/UKM-UKM/2/1). We also thank the research team for their commitment and tireless efforts in producing the manuscript.

The authors acknowledge the Ministry of Higher Education Malaysia and The Universiti Kebangsaan Malaysia, (UKM) for funding this study under the Long-Term Research Grant Scheme-(LGRS/1/2019/UKM-UKM/2/1). We also thank the research team for their commitment and tireless efforts in producing the manuscript.

Additional Editor Comments:

Dear authors, your manuscript has now received comments from the reviewers. You will notice that they are recommending that you make revisions to the manuscript in response to their feedback. We will reevaluate your paper for publication in Plos ONE if you are willing to make revisions to your manuscript. Thanks.

Reviewers' comments:

Reviewer's Responses to Questions

**Comments to the Author**

1. Is the manuscript technically sound, and do the data support the conclusions?

Reviewer #1: Yes

Reviewer #2: Yes

2. Has the statistical analysis been performed appropriately and rigorously? 

Reviewer #1: Yes

Reviewer #2: Yes

3. Have the authors made all data underlying the findings in their manuscript fully available?

Reviewer #1: Yes

Reviewer #2: Yes

4. Is the manuscript presented in an intelligible fashion and written in standard English?

Reviewer #1: Yes

Reviewer #2: No

5. Review Comments to the Author

Reviewer #1: Abstract

This paper examines a convincing issue. Nevertheless, it has a few flaws. Abstracts must provide a concise introduction to the topic, describe how data were gathered, and identify the study site. Kindly explain the research methodology. It must condense the introduction to the issue while expanding the methodology description.

Introduction

Author is required to provide an explanation of the specific state or region in Malaysia that is classified as a drug-based hot spot. What initiatives has the relevant agency undertaken to mitigate the prevalence of drug-related hotspots in Malaysia? What type of incident, such as a drug-related criminal case, is occurring in the hotspot region?

In the article, author describes older adolescents, male, and Western ethnicity are significant predictors of substance abuse on a global scale. How about the demography of Malaysia? Is it identical?

From the present study in Malaysia, what others benefit in studying the factors contributing of drug abuse besides prioritizing of current intervention?

Methodology

Does this research involve face to face interviews? Is that the extent of the researcher's training?

Discussion

The study also reveals that adolescents often exhibit externalizing behaviors such as aggression and rule-breaking, which are strongly associated with an elevated risk of engaging in

drug-use (Fite at al., 2021: Mathias et al., 2015 & Niv et al., 2013).

This statement refers to the male or female in the context of demography?

Conclusion

Rational conclusions are reached. Nevertheless, the challenges associated with cultivating constructive peer networks, equipping parents with information and assistance, carrying out impactful public health campaigns, executing focused interventions, and establishing systems for assessing programmes appear to be widely discussed in the current literature. Which additional policies must be considered within the Malaysian context? In conclusion, evaluate the strengths and weaknesses of this study

Reviewer #2: 1. The introduction already explained the background of study. However, there are only 10 empirical references been referred with repeating the same 3 authors; Brook, Ismail, Nawi. Thus, the author need to add more empirical references.

2. This is a good study, but the article is insufficient to explain the background of study in introduction part. Please explain the theoretical background of the study.

3. The instruments like ASSIST version 3.0 and YSR need to explain more than current (refer the thesis)

4. It is confusing to interpret the results of the sample characteristics and their involvement in drug abuse. The first paragraph shows that the majority of respondents had experience with drug use, but this finding was not consistent with the other reported data. This part needs to be re-examined.

5. The report in drug users by ethnic is hard to understand.

6. PLOS authors have the option to publish the peer review history of their article (what does this mean?). If published, this will include your full peer review and any attached files.

Reviewer #1: No

Reviewer #2: No

---

## [Author Response · Author response to Decision Letter 0]

20 May 2024

REPLY TO THE REVIEWERS. 

BASIC COMMENTS BY REVIEWERS

BIL COMMENT REVIEWER 1 REVIEWER 2 NOTES

1 Is the manuscript technically sound, and do the data support the conclusions?

-The manuscript must describe a technically sound piece of scientific research with data that supports the conclusions. Experiments must have been conducted rigorously, with appropriate controls, replication, and sample sizes. The conclusions must be drawn appropriately based on the data presented.

 YES YES Thank you and we take note.

2 Has the statistical analysis been performed appropriately and rigorously?

 YES YES Thank you.

3 Have the authors made all data underlying the findings in their manuscript fully available?

-The PLOS Data policy requires authors to make all data underlying the findings described in their manuscript fully available without restriction, with rare exception (please refer to the Data Availability Statement in the manuscript PDF file). The data should be provided as part of the manuscript or its supporting information, or deposited to a public repository. For example, in addition to summary statistics, the data points behind means, medians and variance measures should be available. If there are restrictions on publicly sharing data—e.g. participant privacy or use of data from a third party—those must be specified.

 YES YES Thank you.

4 Is the manuscript presented in an intelligible fashion and written in standard English?

-PLOS ONE does not copyedit accepted manuscripts, so the language in submitted articles must be clear, correct, and unambiguous. Any typographical or grammatical errors should be corrected at revision, so please note any specific errors here.

 YES NO Thank you for pointing this out. The article has been edited by a professional editor, however the manuscript has undergone the second revision by an international professional editor. 

DETAILS

CHECK LIST AFTER REVISION

BIL ITEM NOTES

1 A rebuttal letter that responds to each point raised by the academic editor and reviewer(s). You should upload this letter as a separate file labeled 'Response to Reviewers'. A rebuttal letter has been prepared to respond to the reviewers.

2 A marked-up copy of your manuscript that highlights changes made to the original version. You should upload this as a separate file labeled 'Revised Manuscript with Track Changes'. All changes has been highlighted in the revised manuscript

3 An unmarked version of your revised paper without tracked changes. You should upload this as a separate file labeled 'Manuscript'. The unmarked version has been prepared.

JOURNAL REQUIREMENT

BIL ITEM

 NOTES

1 When submitting your revision, we need you to address these additional requirements. Please ensure that your manuscript meets PLOS ONE's style requirements, including those for file naming. The PLOS ONE style templates can be found at https://journals.plos.org/plosone/s/file?id=wjVg/PLOSOne_formatting_sample_main_body.pdf and https://journals.plos.org/plosone/s/file?id=ba62/PLOSOne_formatting_sample_title_authors_affiliations.pdf

 Thank you. The manuscript has been edited to meet the journal’s style requirements. 

2 Thank you for stating the following financial disclosure: The authors acknowledge the Ministry of Higher Education Malaysia and The Universiti Kebangsaan Malaysia, (UKM) for funding this study under the Long-Term Research Grant Scheme-(LGRS/1/2019/UKM-UKM/2/1). We also thank the research team for their commitment and tireless efforts in producing the manuscript.

Please state what role the funders took in the study. If the funders had no role, please state: "The funders had no role in study design, data collection and analysis, decision to publish, or preparation of the manuscript." If this statement is not correct you must amend it as needed. Please include this amended Role of Funder statement in your cover letter; we will change the online submission form on your behalf

 We noted this and have already edited it. 

3 Thank you for stating the following in the Acknowledgments Section of your manuscript: The authors acknowledge the Ministry of Higher Education Malaysia and The Universiti Kebangsaan Malaysia, (UKM) for funding this study under the Long-Term Research Grant Scheme-(LGRS/1/2019/UKM-UKM/2/1). We also thank the research team for their commitment and tireless efforts in producing the manuscript. 

We note that you have provided funding information that is not currently declared in your Funding Statement. However, funding information should not appear in the Acknowledgments section or other areas of your manuscript. We will only publish funding information present in the Funding Statement section of the online submission form. Please remove any funding-related text from the manuscript and let us know how you would like to update your Funding Statement. 

Currently, your Funding Statement reads as follows: The authors acknowledge the Ministry of Higher Education Malaysia and The Universiti Kebangsaan Malaysia, (UKM) for funding this study under the Long-Term Research Grant Scheme-(LGRS/1/2019/UKM-UKM/2/1). We also thank the research team for their commitment and tireless efforts in producing the manuscript. Please include your amended statements within your cover letter; we will change the online submission form on your behalf

 Thank you for your message regarding this matter. We have noted your guidance and have already removed the acknowledgement of funding form the main manuscript as per your instructions.

Our revised Funding Statement is as follows:

‘The authors acknowledge the Ministry of Higher Education Malaysia and The Universiti Kebangsaan Malaysia, (UKM) for funding this study under the Long-Term Research Grant Scheme-(LGRS/1/2019/UKM-UKM/2/1). However, the funders had no role in study design, data collection and analysis, decision to publish, or preparation of the manuscript’ . 

4 We note that your Data Availability Statement is currently as follows: All relevant data are within the manuscript and its Supporting Information files. Please confirm at this time whether or not your submission contains all raw data required to replicate the results of your study. Authors must share the “minimal data set” for their submission. PLOS defines the minimal data set to consist of the data required to replicate all study findings reported in the article, as well as related metadata and methods (https://journals.plos.org/plosone/s/data-availability#loc-minimal-data-set-definition). For example, authors should submit the following data:- The values behind the means, standard deviations and other measures reported;- The values used to build graphs;- The points extracted from images for analysis. Authors do not need to submit their entire data set if only a portion of the data was used in the reported study. If your submission does not contain these data, please either upload them as Supporting Information files or deposit them to a stable, public repository and provide us with the relevant URLs, DOIs, or accession numbers. For a list of recommended repositories, please see https://journals.plos.org/plosone/s/recommended-repositories

 The raw data set has been published in Mendeley Data. Here’s the URL for your reference : https://data.mendeley.com/datasets/ggv2cd88my/2

5 Please include captions for your Supporting Information files at the end of your manuscript, and update any in-text citations to match accordingly. Please see our Supporting Information guidelines for more information: http://journals.plos.org/plosone/s/supporting-information.

 Done, thank you. 

REVIEW COMMENT TO AUTHOR

REVIEWER 1

Comment 1: 

Abtract

This paper examines a convincing issue. Nevertheless, it has a few flaws. Abstracts must provide a concise introduction to the topic, describe how data were gathered, and identify the study site. Kindly explain the research methodology. It must condense the introduction to the issue while expanding the methodology description.

Reply: Thank you for your insightful comments on our manuscripts. We appreciate your acknowledgement of the issue we address in our paper and your constructive comments regarding its presentation. In response to your feedback, we have revised the abstract to ensure it meets the criteria you outlined, we have condensed the introduction to the issue while expanding the description of our research methodology. The adjustment as follow:

For the introduction to the topic, how data were gathered and the study site: The Malaysian Anti-drug Agency (AADK) has identified 155 hotspot areas across Malaysia of which 78 were randomly selected as the study sites. These areas were identified as having extreme drug-related activities such as drug trafficking and drug-related crimes. According to the literature, several factors influence adolescents to be involved in DSA. Therefore, understanding the risk factors in the context of Malaysian school-going adolescents is of utmost importance.

For the methodology, we added: Adolescents in the hotspot areas were selected by means of a cross-sectional survey design with a cluster sampling method. The sample comprised 3382 school-going adolescents, and the data were collected through face-to-face interviews. The logit model method with STATA software was used to analyse the data.

Comment 2:

Introduction:

Author is required to provide an explanation of the specific state or region in Malaysia that is classified as a drug-based hot spot. What initiatives has the relevant agency undertaken to mitigate the prevalence of drug-related hotspots in Malaysia? What type of incident, such as a drug-related criminal case, is occurring in the hotspot region? In the article, author describes older adolescents, male, and Western ethnicity are significant predictors of substance abuse on a global scale. How about the demography of Malaysia? Is it identical? From the present study in Malaysia, what others benefit in studying the factors contributing of drug abuse besides prioritizing of current intervention?

Reply: 

Regarding the identification of drug-based hotspots in Malaysia, we have provided a clear explanation of the specific state or region classified as such. In text, we have added: 

In Malaysia, a total of 155 drug-related hotspots have been identified by the AADK in 2020. This overall number of hotspots was found to have decreased from the total of 178 hotspots reported in 2018. All 155 of these areas encompass all localities in Peninsular Malaysia, including Sabah and Sarawak. The breakdown of the hotspots is as follows: Perlis (6 hotspots), Kedah (13), Pulau Pinang (7), Perak (14), Selangor (13), Federal Territory of Kuala Lumpur (8), Kelantan (15), Terengganu (15), Pahang (12), Melaka (10), Negeri Sembilan (6), Johor (17), Sarawak (7), Sabah (9) and Federal Territory of Labuan (3). These hotspots are focal points identified as grappling with drug addiction issues, with 60% of the entire nation’s population under surveillance, while also serving as regions contributing to social issues such as social tourism and drug-related crimes such as robbery and drug trafficking. The designation of a hotspot is based on four primary indicators: total of individuals under surveillance, total of suspected individuals, total of new addicts and community readiness level. 

Additionally, we have elaborated on the initiatives undertaken by the relevant agencies to mitigate the prevalence of drug-related hotspots in Malaysia. In text, we have explained it as follows: 

One of the strategies to minimize the drug-related activities by 2025 is ‘Hotspots Greening Program’ involving 155 hotspots in Malaysia, implemented collaboratively with government agencies and non-governmental organizations. The program’s strategies are a) preventing new drug addicts, b) maintaining recovery rates, c) increasing drug addict detection, d) enhancing cooperation and involvement of agencies or communities, and e) restricting drug entry. 

To answer to the question ‘What type of incident, such as a drug-related criminal case, is occurring in the hotspot region’, we added:

These hotspots are focal points identified as grappling with drug addiction issues, with 60% of the entire nation’s population under surveillance, while also serving as regions contributing to social issues such as social tourism and drug-related crimes such as robbery and drug trafficking.

For comment ‘In the article, author describes older adolescents, male, and Western ethnicity are significant predictors of substance abuse on a global scale. How about the demography of Malaysia? Is it identical?’, we already confirmed the similarity with additional text as follows:

Globally, males, older adolescents, and Western ethnicity tend to report higher instances of drug use compared to other demographic groups. Similarly in Malaysia, older age groups (Ismail et al., 2022) and male adolescents have been found as more at risk (Hong & Peltzer, 2019; Institute for Public Health, 2017; Institute of Public Health, 2012) than the counterparts. In terms of race, over all Bumiputeras has been reported as the high-risk ethnic group in substance and drug use followed by Indians and Chinese compared to Malays although when it comes to individual substances different these ethnicities differ (Institute for Public Health, 2017; Rodzlan Hasani et al., 2021).

For the comment ‘From the present study in Malaysia, what others benefit in studying the factors contributing of drug abuse besides prioritizing of current intervention?’, we added the explanation as follows: 

It is crucial to focus studies on hotspots to better understand their unique characteristics and the course of action needed to obtain effective preventive treatment. The efficacy of crime prevention of crime prevention strategies is augmented when directed towards identified hotspots and areas exhibiting heightened criminal activity (Weisburd et al., 2016). A crime prevention study focusing on 110 hotspots identified led to a significant reduction in crime within eight months (Koper, 2005).

And, 

The research will elucidate several potential predictors comprising socio-economic factors, peer influences, family dynamics, educational experiences, access to drugs and community characteristics. By considering this broad spectrum of variables, we aim to construct a comprehensive and holistic picture of the factors that contribute to drug abuse among Malaysian school-going adolescents in hot-spot areas. Through in-depth understanding of these predictors, we can develop targeted and context-specific strategies to prevent and address substance abuse in these high-risk communities (Substance Abuse and Mental Health Services Administration US, 2016).

The significance of this research lies in its potential to address several critical issues. Firstly, it enables a nuanced understanding of the challenges faced by adolescents in drug-based hot-spot areas, recognizing that their experiences may differ subtly or significantly from those in other regions. Secondly, it provides a foundation for evidence-based policymaking and intervention design, ensuring that resources are allocated efficiently and effectively to combat substance abuse in the areas where it is most prevalent and crucial. Ultimately, the findings from this study hold the promise of making a substantial contribution to the ongoing efforts to address substance abuse in Malaysia. Understanding the complex interplay of factors in these high-risk communities, we aspire to empower policymakers, educators, and healthcare professionals with the knowledge needed to develop targeted interventions, reduce substance abuse rates, and promote the well-being and prospects of adolescents in these vulnerable 

---

## [Editor Report · Decision Letter 1]

31 May 2024

PLOS ONE Subject: Submission of Manuscript entitled Predictors of Drug and Substance Abuse among School-going Adolescents Living in Drug Hotspot in Malaysia.

PONE-D-23-40678R1

Dear Dr. SHAFURDIN,

We’re pleased to inform you that your manuscript has been judged scientifically suitable for publication and will be formally accepted for publication once it meets all outstanding technical requirements.

Kind regards,

Nik Ahmad Sufian Burhan

Academic Editor

PLOS ONE

Additional Editor Comments (optional):

Dear Dr. Shafurdin. Thank you for revising the manuscript and addressing the reviewers' comments. I'm glad to inform you that your work has been accepted for publication in PLOS ONE. Regards, Burhan